# Reliability of Smart Phone Photographs for School Eye Screening

**DOI:** 10.3390/children9101519

**Published:** 2022-10-04

**Authors:** Rajat M. Srivastava, Suchi Verma, Shubham Gupta, Apjit Kaur, Shally Awasthi, Siddharth Agrawal

**Affiliations:** 1Department of Ophthalmology, King George’s Medical University, Lucknow 226003, India; 2Department of Pediatrics, King George’s Medical University, Lucknow 226003, India

**Keywords:** vision screening, Bruckner’s reflex, school eye screening, smartphone photography

## Abstract

Smartphone photographs capturing Bruckner’s reflex have demonstrated reliability in identifying amblyogenic conditions in children. Assessing visual acuity for screening has been the traditional method since the inception of school screening. The present study aims to assess the reliability of smartphone photographs in detecting ocular morbidities in school children and to compare it with traditional vision screening. Two thousand five hundred and twenty school children underwent vision screening and smartphone cameraphotography by a trained research assistant followed by a comprehensive eye examination of all children by an ophthalmologist. Children with unaided visual acuity less than 6/12 in either of the eyes were graded as abnormal. Based upon the characteristics of the Bruckner’s reflex, the photographs were graded as normal or abnormal by two investigators blinded to the clinical findings. Statistical analysis was performed to compare the sensitivity and specificity of traditional vision screening and photograph based screening, considering comprehensive eye examination as the gold standard. The sensitivity and specificity of vision screening was 81.88% and 97.35% whereas for photographs it was 94.69% and 98.85% respectively. When the two methods were compared, the *p* value was <0.05. We conclude that smartphone photography is better than traditional vision screening for detecting ocular morbidities in school children.

## 1. Introduction

Childhood blindness and visual impairment (VI) is a global concern. Not only does this affect the overall efficiency and productivity of society, they have also been linked to child mortality [1,2]. The World Health Organization (WHO), through the VISION 2020 program highlighted the fact that a large majority of blinding diseases in children are either treatable or preventable and simple measures in the form of vision screening and refractive correction can significantly reduce this burden [1].

A school eye screening (SES) program in India was initiated in the year 1994 under the National Program for Control of Blindness (NPCB) and has since been an integral part of the national health program [3]. Considering the high burden of childhood visual impairment, and limited trained resources in India, school eye screening by ophthalmologists and optometrists has not been a feasible model [3,4,5]. Furthermore, vast and varied terrain also affected the availability of trained professionals in remote areas. Vision screening using trained teachers has been found to be an effective alternative and is a widely used model all over the world [6,7]. However, a lack of standardized protocol for visual acuity assessment, the use of variable optotypes, non-uniform training of teachers, varied awareness and the compliance of children and lack of direct supervision by a trained professional can affect the quality and outcome of school vision screening by teachers [8].

Over the years, the screening process has evolved and photo-screeners with inbuilt algorithms are being propagated in developed countries for efficient screening [9,10]. These are more objective and are considered superior to traditional vision testing especially in younger children [11]. However, due to the prohibitive cost and limited proven efficacy in population-based studies, these are not viable options in a developing country such as India [12]. Recently, GoCheckKids, a smartphone based screening app, has been used to detect amblyopia risk factors in young children. This smartphone application automatically processed the images taken by an iPhone 7 to detect amblyopia risk factors in young children and was found to be a viable alternative to other photoscreeners [13,14]. A recent Indian study by the authors reported the reliability of smartphone photographs for detecting the amblyogenic conditions in children in outpatient department (OPD) settings [15]. In light of encouraging results from the above-mentioned studies, smartphones can become an ideal screening tool in developing countries due to their widespread use and affordable cost. Additionally, smartphone photos may be analyzed by an expert located outside the screening region, allowing for the circumvention of physical and economic barriers to medical treatment, particularly in developing nations. The authors thus conducted the present study to assess the reliability of smartphone photographs for screening school children for ocular morbidities.

## 2. Materials and Methods

This was a cross sectional observation study conducted over a period of 22 months from 2020 to 2022. The study was conducted adhering to the tenets of the Declaration of Helsinki and after obtaining approval from the institutional ethics committee (Ref Code: 90th ECM II A/P 24 dated 20 July 2018). The ‘Basic Shiksha Adhikari’ or ‘Basic Education Officer’ was approached to obtain necessary permission to carry out eye screening of school-going children in government schools. With the assistance of ‘Block Level Education Officers’, a list of various government schools in the region was drafted and the entire region was then subdivided into five different zones—east zone, west zone, north zone, south zone and central zone. Schools were then randomly selected from each zone to ensure representation of schools from each zone. The screening of school children was then performed after taking the permission of the principal of the concerned school.

A sample size of 2500 was calculated using the appropriate statistical variables for community-based screening considering a confidence level of 95%, using formula n = z^2^P (1 − P)/d^2^. (z = 1.96) (*p* = 0.2) (d = 0.0125). On average, about 500 children were planned to be screened from each zone.

Students in the age group between 7 and 15 years from classes 3 to 10 standard were considered for this study. All students from a randomly selected class in a school were subjected to the screening process. Any student who was uncooperative for visual assessment and/or photography was excluded from the study. Each selected student was given a unique alphanumeric identity and this was used for labelling the clinical examination sheet and the digital photographs. The selected students randomly underwent smartphone photography or visual acuity assessment as the first screening procedure. This was done to minimize the occurrence of any systematic bias in the study. The screening process was carried out by a trained research assistant. The research assistant was a science graduate who received 2 months of training by investigators to measure visual acuity and take appropriate photographs with a smart phone. The traditional visual acuity assessment was performed in an adequately lit room of the school using the 6/12 line of the front illuminated Snellen’s E Chart placed at a distance of six meters. The line consisted of five ‘E’ optotypes placed sided by side and each student was asked to identify the optotypes with each eye for vision screening. Both the uncorrected and spectacle-corrected (for children with spectacles) vision for each eye of a student was documented. Monocular vision assessment was ensured by placing an occluder over the other eye. Children with uncorrected visual acuity <6/12 (defined as less than four correct responses out of five) in either eye were categorized as ‘visually impaired’ by the research assistant. This was followed by a comprehensive eye examination comprising of distance visual acuity assessment using a standard illuminated portable Snellen’s box, torch light examination, direct ophthalmoscopy and un-dilated retinoscopy of all the selected students by an ophthalmology senior resident (evaluator). All students with uncorrected distance visual acuity of less than 6/12 in either of the eye or in presence of any ocular morbidity were termed ‘visually impaired’ or ‘abnormal’ by the evaluator. The undilated retinoscopy was performed with a child sitting comfortably and fixating at 6/60 Snellen’s E optotype to minimize accommodative effort, as fogging or the installation of cycloplegic agents was not feasible in a school setting. The findings from the traditional vision screening by the research assistant were compared with that of the comprehensive eye examination by the evaluator (gold standard in this study) to measure its sensitivity and specificity.

All the selected children also underwent smart phone photography targeted to capture the un-dilated red glow (Bruckner’s Reflex) from both eyes by the trained assistant. The photographs were taken in a similar manner to that described by the authors in a study published by Gupta et al. [15]. The photographs were taken in a semi dark room with the child focusing on an illuminated point source of light placed around 6 m in a straight ahead gaze. The smartphone camera lens and flash were placed about 1 m away, parallel to line of sight, to capture the Bruckner’s reflex. All photographs were taken using the primary camera of a smartphone (Model: Vivo S1 Pro manufactured by Vivo Mobile India Private Limited, India) that had a quad camera configuration and cost about twenty thousand rupees (US$270) in 2020 (Figure 1). The camera specifications were 48MP + 8MP+ 2MP +2MP with an f/1.8 aperture and a light emitting diode (LED) flash. The photographs were captured using the default ‘flash-On’ camera setting. No post processing of photographs was performed to alter the Bruckner’s reflex. A total of three consecutive photos capturing red glow from both eyes in each photograph were clicked for each child. Once the photographs were taken, they were labelled with the child’s unique identification number and stored in a hard drive.

All the photographs were then presented to the principal (PI) and co-principal (co-PI) investigators for interpretation. The Bruckner’s reflex was interpreted based upon the color, size and symmetry between the eyes [13]. All three photographs taken for each student were viewed simultaneously, in full screen mode on a 15-inch HP desktop (Model: ProOne 400, manufactured by Hewlett Packard Company, Palo Alto, CA, USA) with standardized (50%) backdrop illumination. These photos were independently interpreted by the PI and co-PI. Both the PI and co-PI were blinded to the clinical assessment findings of the evaluator during their interpretation. Based upon the red glow, the photographs were labelled either ‘normal’, ‘abnormal’ or ‘rejected’ in case the quality was too poor to comment upon the glow. The photographs were labelled abnormal when the abnormality in the glow was observed in at least two out of the three smartphone photographs. Their interpretation was eventually compared with the clinical findings of the evaluator (the gold standard for this study) to assess the sensitivity and specificity of smartphone assisted school screening (Figure 2).

The clinical data and responses of the investigators and research assistant were documented in a Microsoft Excel sheet. The statistical analysis was performed using the Statistical Package for Social Sciences (SPSS) software version 16. An overall analysis of all the data was performed and the qualitative data were presented in numbers and percentages whereas the quantitative data were expressed as mean ± standard deviation. A joint probability of agreement between the investigators was calculated and expressed as percentages for unanimously classified photographs. The photographs of students which were rejected by either of the investigators were excluded while performing sensitivity/specificity analysis of traditional vision screening and smartphone photograph screening to maintain uniformity. The sensitivity, specificity, positive and negative predictive values, expressed as percentages, were calculated both for traditional vision and smartphone photograph screening. Qualitative variables were compared using the Chi-Square test/Fisher’s exact test as appropriate, whereas the independent-t test was applied to compare quantitative variables. A ‘*p*’ value of <0.05 was considered statistically significant.

## 3. Results

### 3.1. Screened Population

A total of 2520 children from 39 government schools were screened using traditional vision and smartphone photograph screening. Out of the 39 schools selected for the study, the majority of them belonged to the north zone (13), followed by the south zone (11) and the central zone (9). Three schools each were selected from the east and the west zones. (Figure 3). The mean age of the screened population was 12.96 ± 2.14 years. The majority of the children were females with an M:F ratio of 1:1.9 (893 males, 1627 females). Due to the poor quality of their photographs, 51 (2.02%) children were excluded from further analysis. The clinical and photographic data of the remaining 2469 (97.98%) students were considered for further analysis. A total of 262 (10.61%) students were found to be visually impaired by the evaluator and suffered with some ocular morbidity. The most common ocular morbidity detected by the evaluator among screened children was refractive error followed by strabismus. Based on dry retinoscopy findings, myopia was diagnosed in 188 (7.4%) and hypermetropia was found in 41 (1.6%) children. The frequency of exotropia (n = 9) was more than twice that of esotropia (n = 4). Sixteen students who were grouped as indeterminate had either more than one ocular morbidity or their exact diagnosis could not be established (Figure 4).

### 3.2. Screening Using Smart Phone Photographs

Besides the rejected photographs, smartphone photographs of 2469 students were screened by the PI and co-PI independently for detecting any ocular morbidity. These were graded as ‘normal’ or ‘abnormal’ in accordance with the previously described parameters mentioned in the methodology [13]. Of these, 2146 photographs were unanimously graded by the investigators demonstrating a joint probability of agreement of 86.91%. (Table 1).

The results of screening using smartphone photographs was compared to the findings of comprehensive clinical examination performed by the evaluator. For unanimously categorized photographs, the sensitivity and specificity values were calculated to be 94.69% and 98.85% respectively. (Figure 5 and Table 2) The positive predictive value and negative predictive value for screening with photographs were 90.67% and 99.37% respectively.

We also analyzed the sensitivity and specificity after including the photographs in which there was disagreement between the investigators. When the photographs with disagreement were categorized as normal, the sensitivity and specificity were calculated to be 81.67% and 99% respectively, whereas, when these photographs were considered as abnormal, the sensitivity and specificity values were 95.41% and 86% respectively (Table 3 and Table 4). Thus, the sensitivity of photographs for identifying the presence of ocular morbidity in our study ranged between 81.67% and 95.41% whereas the specificity ranged between 86% and 99%. Of the photographs rejected for analysis, only 7/51 (13.7%) children were found to be suffering from visual impairment on clinical examination.

### 3.3. Traditional Vision Screening

Vision screening was performed by the research assistant using the 6/12 line on the Snellen’s E chart. Out of the 2469 school children, 284 were found to have visual impairment. The sensitivity and specificity for traditional vision screening in our study was 81.88% and 97.35% respectively. (Table 5).

### 3.4. Smart Phone Screening vs. Traditional Vision Screening

The sensitivity and specificity of screening with photographs in our study was found to range between 81.67% and 95.41% and 86% and 99% respectively. However, the sensitivity and specificity for photographs was calculated to be 94.60% and 98.85% when the interpretation of photographs was unanimous. This was better compared to the sensitivity (*p*-value is <0.001) and specificity (*p*-value is <0.001) of traditional vision screening (Table 6).

## 4. Discussion

The reliability of smartphone photographs for detecting amblyogenic conditions in children under controlled OPD settings has already been demonstrated in the past by the authors [15]. The present study is among the first and largest community-based study from India to assess its reliability for screening ocular morbidities in school children. The study also compares the method of traditional vision screening with smart phone photographs-based screening of school children.

Both the acquisition of smartphone photographs and vision screening in the present study were carried out by a trained research assistant. The research assistant was a science graduate who received 2 months of training by investigators to measure visual acuity and take appropriate photographs with a smart phone. Our study demonstrated that screening with smartphone photographs was better compared to the method of traditional vision screening. Past studies have demonstrated that the outcomes of traditional vision screening are often affected by the training and motivation of the screening personnel [16]. This is also highlighted in the present study as the sensitivity and specificity of traditional vision screening in our study by a trained research assistant was better than those reported by school teachers [7,17]. However, factors such as the distance from which the vision is checked, the lighting conditions of the room, the quality of Snellen’s chart, the size of the optotype used etc. can compromise the outcome even in the presence of trained staff. The use of the 6/12 Snellen optotype for vision screening in the present study has also improved the specificity of screening without compromising the sensitivity, similar to what has been demonstrated in the previous study by Saxena et al. [18]. The scope of vision assessment as a screening tool is also limited due to the inherent subjectivity arising due to the variable skill of the examiner and the understanding of the child.

The idea to screen for ocular morbidities using Bruckner’s reflex is not new and various studies have attempted to study its efficacy [19,20,21]. Photographic analysis of Bruckner’s reflex was attempted by Kakkinen K et al. way back in 1981 [22,23]. The authors described the fundal glow in different refractive errors and proposed its clinical utility in diagnosing strabismus and high refractive errors. The principle was put to the test in India by Raza SA et al. who demonstrated the utility of the Canon CP-TX1 camera as a screening tool for amblyogenic risk factors in 262 Indian children. They reported a sensitivity and specificity of 82% and 98% for the detection of amblyogenic risk factors in OPD-based settings. [24]. A recent population-based study from India involving young children using the Plusoptix S12-C photoscreener for detecting amblyogenic risk factors demonstrated a sensitivity and specificity of 86.76% and 82.27% respectively [25]. The study screened 367 children in the age group between 6 months and six years. Both these studies required special equipment for carrying out the screening process.

The present study is the largest population based study involving more than 2400 school children who were screened for visual impairment using a smartphone. The sensitivity and specificity for detecting visual impairment using smartphone photographs in the present study have ranged between 81.67% and 95.41% and 86% and 99%. When the photographs were interpreted unanimously, 94.69% and 98.85% sensitivity and specificity could be achieved, which is similar to that reported by the authors in a previous hospital-based study [15]. The results in the present study are also comparable to the sensitivity and specificity reported by Arnold RW et al. in their study on detecting refractive amblyogenic risk factors with manual grading of photographs taken using an iPhone 7 [13]. This level of accuracy in our study is attributable to a uniform interpretation of standardized photographs by experienced ophthalmologists. Screening with photographs in the present study also exhibited 100% accuracy in the identification of strabismus, ptosis and congenital ocular anomalies.

While screening with photographs may also help identify ocular morbidities other than refractive error, it is difficult to accurately quantify refractive errors based on the present methodology. Artificial and deep machine learning devices such as ‘Kanna’ photoscreeners and smartphone applications such as GoCheckKids have shown promising results in the identification and quantification of refractive errors [13,26]. Demonstrable accuracy achieved with smartphone photography based screening in the present study may pave the way for enhancing the quality of ocular morbidity screening in children by using standardized machine based algorithms. Further large community-based studies may be conducted to develop and improve such algorithms.

While interpreting photographs, there was a disagreement between investigators in the interpretation of 323 (13%) photographs. Despite a high level of agreement between the investigators, differences arising out of intrinsic subjectivity between investigators cannot be ruled out. This may be expressed as disagreement, especially when interpreting large samples of photographs. This is evident as the investigators exhibited almost 100 percent agreement in a previous hospital-based study performed involving only 250 children. Disagreement between investigators was most evident in photographs which were overexposed, slightly defocused or where the subject was not looking straight ahead. It was observed that deviation from standard technique as described in the methodology may lead to more subjective interpretations and high errors. As described earlier, poor photographs prevented the screening of only 2% of enrolled children which is significantly less than the 11% cited in another smartphone photograph-based study [13]. Failure rates similar to those in the present study have been experienced even with traditional vison screening studies [27].

Ocular morbidity was detected in 10.4% of children screened in the present study, similar to as reported by Nepal BP et al. in Nepali school children [28]. The occurrence was higher compared to what has been reported by other Indian studies as vision screening in the present study was performed by a well-trained research assistant and not by schoolteachers [17]. The majority of screen positive children suffered from refractive errors (87.4%) followed by strabismus (5%). The overall prevalence of refractive error based upon dry retinoscopy findings in children was 9%. Myopia was much more common than hypermetropia and exotropia was twice as common as esotropia, similar to what has been observed in Nepali school children [28]. Only 1.6% (41) of children were diagnosed with Hypermetropia compared to 7.7% by Murthy GV et al. [29]. Mild to moderate hypermetropia may have been underreported in the present study due to inadequate cycloplegia while performing a retinoscopy. Furthermore, the prevalence of refractive errors is known to have geographic and ethnic variations. The varied ethnicity, age group of participants and inherent differences in the study design could explain such differences in the occurrence of refractive errors in the present study compared to in the western world [30].

The mean age of children in our study is greater than that in the other reports [27]. Older children are more co-operative and this could possibly be a reason for higher accuracy in our study using either of the screening methodologies. It would be interesting to compare the accuracy of smartphone photograph-based screening among a younger population.

Despite being a large cohort, the frequency of ocular morbidities other than refractive errors and strabismus was small. However, the present study provides a unique opportunity for a realistic portrayal of the occurrence of ocular morbidities in school children. A lack of bias in reporting by virtue of its community-based designs adds to the strengths of the study.

One of the major limitations of our study was the non-quantification of refractive errors in screened children. A handheld auto-refractometer would have been a useful device for measuring refractive errors and further analysis. Furthermore, the criteria for the detection of visual impairment by a digital screening tool were not in accordance with the AAPOS guidelines thereby limiting the comparison with other mentioned studies [31]. The criteria for the interpretation of photographs warrants further refining to reduce disagreement. This would be necessary for conducting larger population-based screening in the future. Additionally, the reproducibility of photographs demonstrating Bruckner’s reflex in the inexperienced hands of schoolteachers is yet to be determined before the proposed method can be widely accepted as an effective and affordable screening modality for school eye screening programs in developing countries. Collecting and analyzing larger data with varied morbidities will enable the objectification of the photograph interpretation including the quality of glow. This can be developed into a mobile application for school eye screening.

## 5. Conclusions

Smart phone photograph screening is a good alternative to the traditional vision screening of school children. Interrater agreement for interpretation is highest in the presence of good quality photographs. The sensitivity and specificity of screening with photographs (94.69% and 98.85) is better compared to traditional screening vision screening (81.88% and 97.35%), *p* < 0.05.

## Figures and Tables

**Figure 1 children-09-01519-f001:**
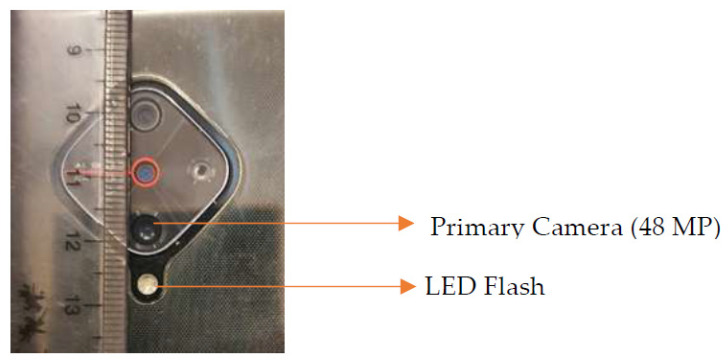
Photograph demonstrating the arrangement of primary camera lens and ‘LED’ flash on the smart phone.

**Figure 2 children-09-01519-f002:**
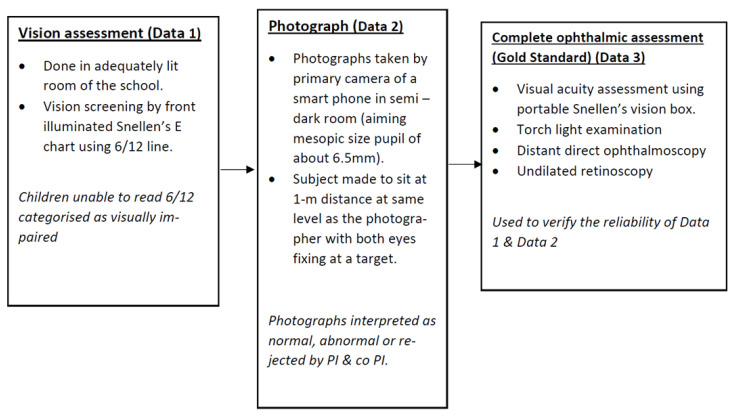
Flowchart depicting the scheme of events in the study. Datasets 1 & 2 were compared with dataset 3. Students randomly underwent visual acuity screening (**Data 1**) or smartphone photography (**Data 2**) as the first screening procedure.

**Figure 3 children-09-01519-f003:**
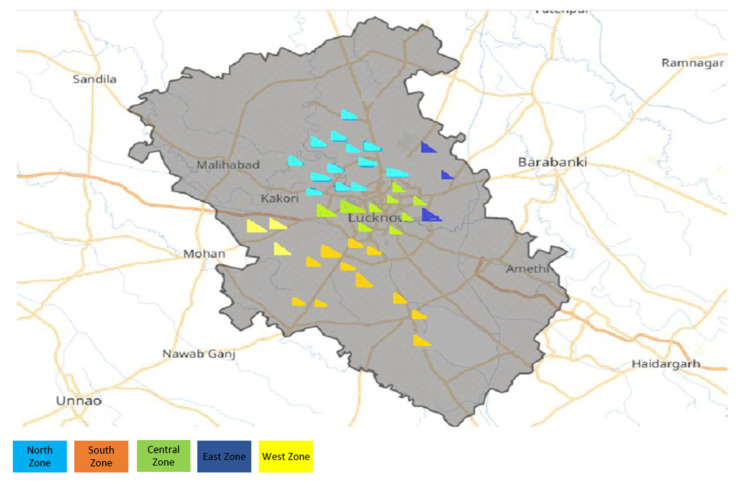
Map depicting zonal distribution of schools screened in the study.

**Figure 4 children-09-01519-f004:**
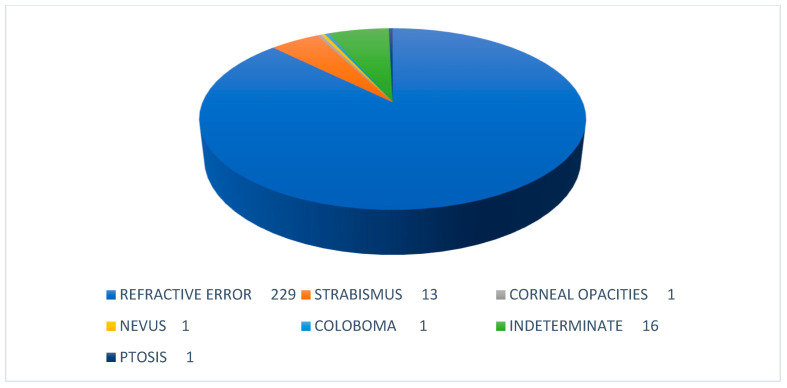
Pie diagram depicting the various ocular morbidities detected in screened population by the evaluator.

**Figure 5 children-09-01519-f005:**
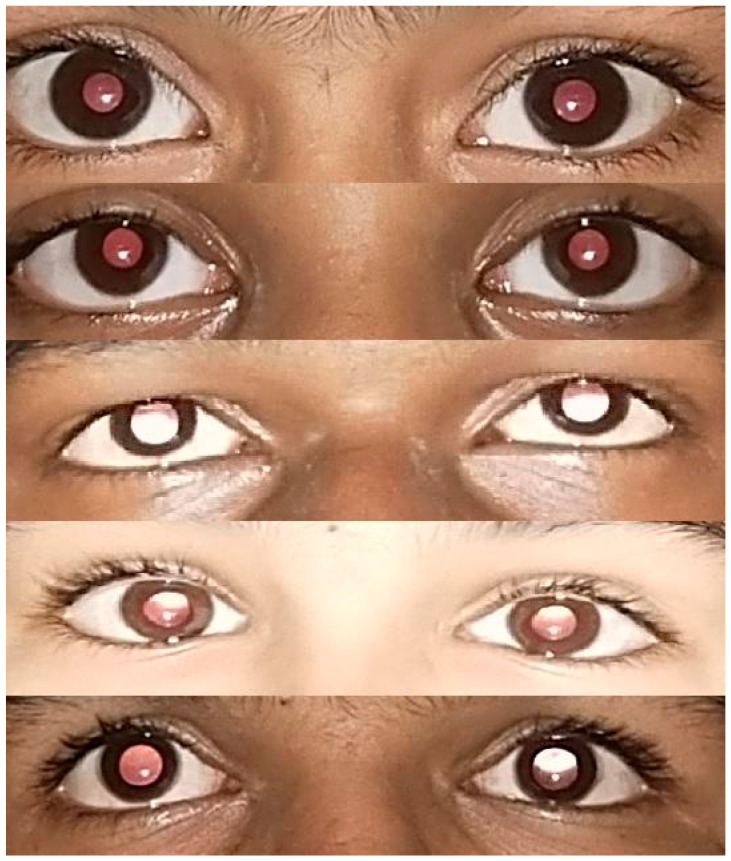
Representative photographs from the sample (top to bottom) 1. Emmetropia 2. Emmetropia 3. Myopia (RE − 4.5 DS, LE − 3.5 DS) 4. Hypermetropia (RE + 2.0 DS/+1.0 DCyl @ 70, LE + 2.5 DS/+1.0 DCyl @ 110) 5. Anisohypermetropia (RE + 0.5 DS, LE + 4.0 DSph/+0.75 DCyl @ 90). (RE = right eye, LE = left eye, DS = Dipotre Sphere, DCyl = Dioptre Cylinder).

**Table 1 children-09-01519-t001:** Screening of smartphone photographs by the investigators.

	NORMAL	ABNORMAL	TOTAL (%)
PI	2135	334	2469 (100)
CO PI	2008	461	2469 (100)
PI + CO PI	1910	236	2146 (86.91)

**Table 2 children-09-01519-t002:** Sensitivity and specificity of detecting ocular morbidities using unanimously graded smartphone photographs. Figures in italics represent the sensitivity and specificity.

Investigators (Smart Phone Photographs)	Evaluator (Clinical Examination)	
	Abnormal (%)	Normal (%)
**Abnormal**	*214 (94.69)*	22 (1.14)
**Normal**	12 (5.30)	*1898 (98.85)*
**TOTAL**	226 (100)	1920 (100)

**Table 3 children-09-01519-t003:** Sensitivity and specificity considering photographs with disagreement as normal.

Investigators (Smart Phone Photographs)	Evaluator (Clinical Examination)	
	Abnormal (%)	Normal (%)
Abnormal	*214 (81.67)*	22 (1)
Normal	48 (18.33)	*2185 (99)*
TOTAL	262 (100)	2207 (100)

**Table 4 children-09-01519-t004:** Sensitivity and specificity considering photographs with disagreement as abnormal.

Investigators (Smart Phone Photographs)	Evaluator (Clinical Examination)	
	Abnormal (%)	Normal (%)
Abnormal	*250 (95.4)*	309 (14)
Normal	12 (4.6)	*1898 (86)*
TOTAL	262 (100)	2207 (100)

**Table 5 children-09-01519-t005:** Sensitivity and specificity of traditional vision screening. Figures in *italics* represent the sensitivity and specificity.

Vision Screening (6/12 Optotype)	Evaluator (Clinical Examination)	
	Abnormal (%)	Normal (%)
Abnormal (<6/12)	*226 (81.88)*	58 (2.64)
Normal (≥6/12)	50 (18.11)	*2135 (97.35)*
TOTAL	276 (100%)	2193 (100%)

**Table 6 children-09-01519-t006:** Sensitivity and specificity of smartphone photographs vs. traditional vision screening.

Screening Modality	Sensitivity	*p*-Value	Specificity	*p*-Value
Photographs	94.69% (81.67–95.41%)	<0.001	98.85% (86–99%)	<0.001
Vision Screening	81.88%		97.35%	

## Data Availability

The photographs and MS-Excel sheets containing the entire data can be made available to the reviewers if required.

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
