# Peer review of "Reliability of Smart Phone Photographs for School Eye Screening"

_children, 2022, doi:10.3390/children9101519_

Round 1
Reviewer 1 Report
This is a very interesting study on a topic which is very important, especially in this setting. Please see below my comments:
The English grammar, spelling and punctuation needs some attention throughout the paper.
Methods:
- can you define what students from class 3-10 means? Could you perhaps give an age range to help an international audience understand?
- what does uncooperative mean? did they refuse screening or were they too difficult to screen?
- 'uncorrected distant vision' - should this be 'uncorrected distance vision'
Results:
It may be beneficial to move figure 2 up to where you first mention zones
3.3 traditional screening
the sentence: '284 school children were screened to have visual impairment' - should this be '284 children who were screened were found to have visual impairment?
Discussion:
Pg 7: the information on the research assistant might be better in the methodology section
Author Response
Dear Reviewer,
Thank you for the review. We have made a point-by-point clarification of the queries raised by you in the attached MS Word File. We have made some changes in the manuscript as well.

Reviewer 2 Report
Summary of Paper: IRB-approved population study, during COVID, with school principal consent of target 2500 (actual; 2520) students class 3-10 from 39 government schools comparing conventional 6/12 Tumbling E acuity screen with smart phone photographs with resident ophthalmologist follow up exams on all with uncorrected 6/18 considered “impaired.”. Gupta-method, Vivo S1 Pro smart phone from 1 meter parallel to distant target fixation taken by trained technician not a teacher – 3 images each were scored by two expert readers blinded to exam outcome. The camera hs high 48MP resolution and an LED “flash” – but whether any “red reflex” reduction option was activated or in-activated not mentioned. The dimensions of the flash and lens were not given.
Age 13±2 years, 65% female. 51 excluded due to poor quality photos and not entered into i=refer extra calculation. 2469 analyzed with 262 having “visual impairment” mainly refractive error (229) or strabismus (13), corneal opacity 1, coloboma 1, ptosis 1, nevus 1, “indeterminate” 16. Grouped photo interpretation had sensitivity 94.7% and specificity 98.9% but the agreement between readers was not perfect with over 300 students conflicting. Phone image exposure and fixation issues mentioned for the disagreement.. Tumbling E acuity had sensitivity 81.9% and specificity 97.4%.
Major Comments:
Screening should target a condition with a latent, phase for which treatment is better early than late (Wilson J, Junger G. Principles and practice of screening for disease. Public Health paper No 34. Geneva: World Health Organization; 1968.) Amblyopia is such a condition in children but the latent phase is the first decade. Refractive error could be a target condition with respect to school performance. Glaucoma, iritis, retinal tumors, intermittent strabismus or cranial nerve palsy are either too rare, or of questionable school intervention to reach a community standard for screening. Why did you start your screening in older children? Please explain.
Other than lack of cycloplegia (which is school-disrupting in older students), you have a robust study model that appear more than a “pilot.” On the other hand, a uniform definition of target conditions (like AAPOS 2021) is needed before validation (sensitivity and specificity) can be compared between two studies. It is OK to make your own definition, you just cannot compare values. Please explain.
The use of the smart phone for vision screening needs to include some references and comments about GoCheck Kids:
Walker M, Duvall A, Daniels M, Doan M, Edmondson LE, Cheeseman EW, et al. Effectiveness of the iPhone GoCheck Kids smartphone vision screener in detecting amblyopia risk factors. J AAPOS. 2020;24(1):16 e1- e5.
Peterseim MMW, Rhodes RS, Patel RN, Wilson ME, Edmondson LE, Logan SA, et al. Effectiveness of the GoCheck Kids Vision Screener in Detecting Amblyopia Risk Factors. Am J Ophthalmol. 2018;187:87-91.
Arnold RW, O'Neil JW, Cooper KL, Silbert DI, Donahue SP. Evaluation of a smartphone photoscreener app to detect refractive amblyopia risk factors in children 1-6 years. Clin Ophthalmol. 2018;12:1533-7.
(Nokia Xenon flash: Arnold RW, Arnold AW, Hunt-Smith TT, Grendahl RL, Winkle RK. The Positive Predictive Value of Smartphone Photoscreening in Pediatric Practices. J Pediatr Ophthalmol Strabismus. 2018;55(6):393-6.)
Martin SJ, Htoo HE, Hser N, Arnold RW. Performance of two photoscreeners enhanced by protective containers. Clin Ophthalmol. 2020;14:1427-35.
Your 10% vision impairment in this group seems low. From routine USA multi-ethnic children, the prevalence of high refractive error and strabismus is about 21%. In older students, the prevalence of myopia is expected to increase. Please comment.
(Arnold RW. Amblyopia risk factor prevalence. J Pediatr Ophthalmol Strabismus. 2013;50(4):213-7.
Varma R. Amblyopia refractive risk factors (letter reply). Ophthalmol. 2012;119(6):1283-4.)
This is a very large study with manifest refractions in a large cohort of students not selected by symptoms of vision loss. A report of the medians, ranges of refractive errors would be very helpful- perhaps in another table.
With the Snellen E charts, how was monocular testing assured?
With your un-dilated retinoscopy, were methods uniformly applied to reduce accommodation? Free lenses, skiascopy or phoropter with fogging?
Validation should be described as if the “rejected” photographs were a “refer.”
In your large group, did any children have “hysterical denial of vision?”
In section 3.4, are you still targeting the exam outcome, of are you targetting visual acuity results from the smart phone photos? Please clarify.
Rather than give a range of mixed interpretation for the two viwers, calculate their “agreement” and report exact results from PI and for CO PI. Without an automated process, it is unlikely that you would be able to have two experts view each image in any practical clincal extension of your design.
Given the 262 students you identified by exam and targetted by screening, how many needed an treatment? How may would have been identified by the school question, “Do you have trouble with your vision?” How many might have been detected by placing a small card with optotype on the board, and asking if they had trouble reading it?
Please give more details about both screening methods. For instance, what were the dimensions of the camera? Was there an accommodative or non-accomodative target for fixation on the camera? Did the camera have a red-eye reduction pre-flash that required over-ride? Did the camera have “Lidar” or other method to focus in the dark?
For visual acuity testing, was the Tumbling E just a “critical line?” Was it surrouned or crowded to select amblyopia? Were optotypes randomly presented, or in a uniform row? Were sutdents able to overhear the prior student’s answers before they were tested? How was monocular testing insured? Did you test both-eyes open for any who failed monocular visual acuity? Was there any orientation phase to make sure the student understood the optotype? How many opotypes correct were required to pass the 6/12 line?
Can you resent a test, re-test result of a subset of your typical students to determine whether actual vision imapired has a (?3%) chance of guessing or a normal child has a (? 5%) chance of failing the visual acuity?
If a student had spectacles and saw well with them, were they included in the “vision impariement” group? Can you report the proportion of children who already had spectacles/
In your discussion, you mention Brückner test and photoscreening papers that reported sensitivites and specificities- targeting very different exam outcome than your study. It is imperative to ALSO report your study re-analyzing using Uniform outcome data even though you did not use cycloplegia- as the 2003, 2013 and 2021 AAPOS guidelines recommend. Since you are streening much older children past their amblyopia age range, I think is acceptable to practically use manifest refraction, but you should mention how hyperopic students had their accommodation relaxed during retinoscopy.
Of the unaccepted photos, what proportion had “vision imapirment” by your exam?
Please describe the inherent limitation of using LED on smart phone compared to actual Xenon flash as initially utilized by GoCheck Kids with Nokia versus iPhone.
The following references deserve review and consideration of comment in your discussion:
Smart phone photoscreening was first mentioned here: (Arnold RW, Davis B, Arnold LE, Rowe KS, Davis JM. Calibration and validation of nine objective vision screeners with contact lens-induced anisometropia. J Pediatr Ophthalmol Strabismus. 2013;50(3):184-90.)
Consumer inexpensive, digital flash camera photoscreening early mentioned here:
(Arnold RW, Arnold AW, Stark L, Arnold KK, Leman RE, Armitage MD. Amblyopia detection by camera (ADBC): Gateway to portable, inexpensive, vision screening. Alaska Med. 2004;46(3):63-72.)
(Kovtoun TA, Arnold RW. Calibration of photoscreeners for threshold contact- induced hyperopic anisometropia: Introduction of the JVC photoscreeners. JPOS. 2004;41(3):150-8.)
A different digital flash camera used recently in India should also be referenced: (Raza SA, Amitava AK, Gupta Y, Afzal K, Kauser F, Saxena J, et al. Canon CP-TX1 camera - As a screening tool for amblyogenic risk factors. Indian journal of ophthalmology. 2022;70(4):1313-6. doi: 10.4103/ijo.IJO_2161_21. PubMed PMID: 35326044; PubMed Central PMCID: PMC9240504.)
Early consumer camera digital photoscreening was compared to patched HOTV acuity in school children here:
(Leman RE, Clausen MM, Bates J, Stark L, Arnold KK, Arnold RW. A comparison of patched HOTV visual acuity and photoscreening. J Sch Nurs. 2006;22(4):237-43.)
Minor Comments:
Table 4 needs another “parenthesis” down at 22070)
Both photoscreening and eye chart are “vision screening” so you might wich to designate the tumbling E chart as “visual acuity screening.”
Discussion paragraph starting with While interpreting photographs… near the end, do you mean categorize?
Author Response
Dear Reviewer,
Thank you for the detailed review which has helped us to improve the manuscript. We have made a point-by-point clarification to the queries raised by you in the attached MS Word file.

Round 2
Reviewer 2 Report
Revision has addressed many of my questions but I was not allowed to see your responses to other reeviewer and other important questions remain unanswered.
Thank you for ytour description of details of the phone which are very important
As for camera qualities, the phone is said to have a “Proximity sensor” which may aid in focus in dim light. (Website for VIVO S1 pro no longer for sale)
The image of the camera lenses with flash and ruler are excellent.
The quad lenses- did they produce four individual images with each flash- or was only one 48 MP image produced. Can you label the power of each lens or the megapixels it produced on your photo of the lenses? Please include these with an example of a quality red reflex photo from the camera showing refractive error and report the amount of exam refractive error for that patient. (Parent consent needed.)
In your description of the tumbling E visual acuity screening and the follow up exam, if patchging was not used, and if an occluder was not used to insure monocularity- please state this. Or state what was uniformly used.
It appears your method to reduce accommodation during retinoscopy was to have the studetn gaze at a 6/60 optotype? If the student was not fogged with higher plus over the non-refracted eye- please state this. You correctly meintioned that this was manifest refraction. Include a statement: accommodation was reduced by ( gaze at a 6/60 optotype) but not by fogging or cycloplegic drops.
Figure 1- compared traditional, Photograph and complete ophthalmic assessment I cannot read fully the outcomes for the middle box (Photograph)
Please include in the third box (Complete ophthalmic assessment) a “True finding” is any student uncorrected worse than 6/12 or any ocular morbidity. (NOT AAPOS UNIFORM GUIDELINE)
Since you have not re-addressed your data using a uniform guideline, please include that in you “Weaknesses of this study include…
Despite having exam outcomes from a large under of school children including manifest refraction, you have not presented helpful, regionally specific demographis- please list this a a weakness of the current study- or mention if that data will be reported subsequently.
You mention a weakness as not having higher proportion of morbidities for the camera to view, but yours is actually a better cohort without enhanced pathology and therfore worthy of describing that at a strength. You could report that strength making this a much better paper.
A uniform definition of visual acuity deficit with ocular mobidities would enhacne comparison be tween yours and reference 28. The lower prevalence of disorders, including defined ranges of refractive errors compared to MEPEDS would help that discussion (Varma R. Amblyopia refractive risk factors (letter reply). Ophthalmol. 2012;119(6):1283-4.)
I’m sorry to see that you might not have reference software like Endotes to make re-writing and adding references simpler.
I believe you have the right to exclude the “Pilot Study” from your Title.
Author Response
Dear Reviewer,
Your valuable suggestions have been included in the manuscript. Thank You
